# Does Catheter Insertion Site Matter? Contamination of Peripheral Intravenous Catheters during Dental Scaling in Dogs

**DOI:** 10.3390/vetsci11090407

**Published:** 2024-09-03

**Authors:** Ivana Calice, Panagiotis Ballas, Claus Vogl, Sandra Purwin, Monika Ehling-Schulz, Attilio Rocchi

**Affiliations:** 1Clinical Unit of Anaesthesiology and Perioperative Intensive-Care Medicine, Clinic for Small Animals, Department for Small Animals and Horses, Veterinary Medicine University of Vienna, 1210 Vienna, Austriaattilio.rocchi@vetmeduni.ac.at (A.R.); 2Institute for Microbiology, Department of Pathobiology, Veterinary Medicine University of Vienna, 1210 Vienna, Austriamonika.ehling-schulz@vetmeduni.ac.at (M.E.-S.); 3Department of Biomedical Sciences, Veterinary Medicine University of Vienna, 1210 Vienna, Austria; claus.vogl@vetmeduni.ac.at

**Keywords:** contamination, peripheral intravenous catheter, dental scaling, dog, cannula

## Abstract

**Simple Summary:**

This study compared the bacterial contamination between two common peripheral intravenous catheter placement sites, in the front leg versus the hind leg, in dogs undergoing dental scaling under general anaesthesia. Regardless of the dog’s size, a significantly higher bacterial contamination load was found at the front leg compared to the hind leg placement site. The bacteria isolated from the injection port of the cannula were mainly Staphylococcus, Neisseria, and Bacillus species. These findings suggest that to reduce the risk of contamination of the peripheral intravenous catheter during dental scaling procedures in dogs, placing the catheter in the hind leg may be beneficial.

**Abstract:**

During dental scaling in dogs under general anaesthesia, contamination of the peripheral intravenous catheter (PIVC) is unavoidable due to splatter and the generated aerosol. Bacterial contamination was compared between two commonly used PIVC placement sites. Thirty-nine client-owned dogs with a minimum length from their nose to their tail base of 50 cm were randomly assigned to receive a PIVC in either their cephalic or saphenous vein. Irrespective of the PIVC placement site, brain heart infusion agar dishes were placed in the cephalic and saphenous vein areas. Their lids were closed 0, 5, and 10 min into the procedure. Contamination was measured by counting the colony-forming units after incubation on different substrates. The data were analysed with descriptive statistics, ANOVA, and ANCOVA (*p* < 0.05). The cephalic vein area showed a significantly higher bacterial load than the saphenous vein area (*p* ≈ 0.0) regardless of the length of the dog. Furthermore, the dorsal PIVC injection ports were sampled before and after scaling, and the colonies isolated were counted and subjected to MALDI-TOF-MS for identification. The bacteria mainly belonged to the genera Staphylococcus, Neisseria, and Bacillus. Our results suggest that for dental scaling in dogs, the PIVC should be placed in the pelvic limb whenever possible to reduce the potential risk of contamination.

## 1. Introduction

During dental scaling, both the operator and the patient are continuously exposed to aerosol generated by use of ultrasonic scaling devices, water, and air turbines [1,2]. The composition of the aerosol differs from patient to patient and according to the nature of the procedure. The large and small air droplets released during the procedure contain bacteria from the teeth and oral cavity, saliva, blood, and coolant fluid [3]. The splatter and aerosol generated during dental scaling procedures can spread to various surfaces, including tools, equipment, and the general environment.

Dental scaling in dogs is performed under general anaesthesia in lateral or dorsal recumbency with the head of the patient turned toward the operator. A peripheral intravenous catheter (PIVC) is placed as part of general anaesthesia management. The contaminants generated during the procedure can therefore also reach the patient’s PIVC since their front and hind limbs are exposed to the aerosol.

Several human studies have investigated the airborne contamination induced by dental scaling by positioning sedimentation plates in the vicinity of the patient. These studies have shown that the distance between and the positioning of the operator and the assistant play a role in the pattern of aerosol contamination. These studies have predominantly compared the contamination pattern of different suctioning systems [1,2]. In veterinary medicine, these systems are, however, rarely available.

Furthermore, the veterinary literature on this topic is sparse, and no data exist on the influence of dental scaling and related aerosol production on PIVC contamination.

The reported prevalence of the microbial colonisation of PIVCs in hospitalised veterinary patients ranges between 10.4% and 39.6% [4,5,6,7,8,9]. Nevertheless, veterinary medicine currently lacks comprehensive guidelines for the best PIVC placement site, which could help reduce contamination and improve patient outcomes.

The primary aim of this study is to compare the contamination of two common PIVC placement sites during dental scaling in anaesthetised dogs using sedimentation plates in close vicinity to the cephalic vein (CV) and the external saphenous vein (SV). We tested the hypothesis that the contamination differs between the CV and SV areas and is influenced by the distance from the oral cavity. A secondary aim was to identify and evaluate potential risk factors influencing the contamination of the PIVC.

## 2. Materials and Methods

### 2.1. Animal Selection

The study protocol was approved by the Institutional Ethics Committee of Vetmeduni Vienna, Austria (ETK-32/06/2018, ETK 18/07/2017). Fifty-one client-owned dogs presenting for an elective dental procedure were recruited. Informed owner consent was obtained for each dog. The inclusion criteria were an ASA classification I to III and a length of the animal from its nose to base of its tail of at least 50 cm. Dogs were excluded if the PIVC could not be placed during a single attempt, i.e., with a single puncture of the skin.

### 2.2. Study Design and Treatments

This study was designed as a prospective randomised clinical trial and separated into two parts. Part A of the study investigated the contamination of the area of the two common PIVC placement sites irrespective of the catheter placement site, and part B looked at the contamination of the dorsal PIVC injection port depending on its placement in either the cephalic vein (CV) or the saphenous vein (SV). Dogs were randomly allocated into the two groups (the CV and SV groups) using a list randomiser (Random.org Randomness and Integrity Services Ltd., Dublin, Ireland. Available at www.random.org/lists; accessed on 3 November 2017). The anaesthetic and dental procedures were performed by the attending anaesthetist and dentist. Two investigators aware of the group allocation (I.C. and S.P.) carried out the instrumentation and performed the data collection. Two other investigators processed the microbiology samples (P.B. and S.P.).

#### PIVC Placement and Anaesthetic and Dental Procedures

To minimise aerosol cross-contamination, the dental scaling procedures were scheduled for the early morning, and no other dental procedures were conducted in the dental theatre for at least 48 h before the experiments. During the experimental part of the procedure, the animal’s position remained unchanged.

A region of hair around the vein of approximately 5 cm × 4 cm was clipped at the insertion site (the middle third of the antebrachium for the CV and the lateral distal surface of the tibial region for the SV). The skin was scrubbed with a cellulose swab soaked in a propanol-based disinfectant (Cutasept F propane–2–ol, Bode Chemie GmbH, Hamburg, Germany). Afterwards, the same disinfectant was sprayed over the area and left to dry for 15 s before the PIVC’s placement. An over-the-needle PIVC with a second dorsal PIVC injection port (Vasofix Safety, Shielded IV catheter with an injection port, 20–22 gauge; B. Braun, Kronberg, Germany) was placed in the CV or the SV depending on the group allocation. The size of the PIVC (20 or 22 G) was chosen according to the size of the vein. Before placing the PIVC, the investigators washed, disinfected (Lifosan soft, and Visco Rub, B. Braun, Melsungen, Germany), and dried their hands and wore non-sterile, single-use gloves (Vasco Nitrile white, B. Braun, Melsungen, Germany). Upon the PIVC’s insertion, a cellulose swab (Pur-Zellin, Hartmann, Heidenheim, Germany) was placed between the skin of the patient and the body of the PIVC. Thereafter, the PIVC was secured onto the limb with medical tape (3M, Durapore, 3M, Neuss, Germany) by an investigator wearing clean gloves.

The anaesthetic drug regime was chosen at the anaesthetist’s discretion. Less cooperative patients were premedicated intramuscularly before the PIVC placement, while the other subjects were premedicated intravenously. Anaesthesia was induced with injectable agents and maintained with isoflurane in oxygen. A balanced crystalloid solution (Sterofundin Iso Infusionsloesung, B. Braun, Melsungen, Germany) was delivered through an extension line (30 cm extension tubing, B. Braun, Melsungen, Germany) connected to the distal PIVC hub, which was not manipulated or sampled further. Additional drugs were administered only through the capped dorsal PIVC injection port. This capped dorsal PIVC injection port was flushed with sterile saline solution (NaCl 0.9%, B. Braun, Germany) (1–2 mL) after each injection. The distances from the nose to the location where the PIVC was placed (either the CV or SV), or would have been placed for the alternative location, were also recorded before the start of the experiment in lateral recumbency (Figure 1).

The severity of periodontal disease was scored as mild, moderate, or severe by the attending dental surgeon, as described elsewhere [10]. Chlorhexidine (Paroex, Sunstar, Etoy, Switzerland) flushing of the oral cavity was performed at least 20 min before the start of scaling, which was conducted as described elsewhere [10,11].

### 2.3. Sampling Procedures

#### 2.3.1. Part A: Sedimentation Plates—Sampling Procedures

Sedimentation agar plates (Thermofischer, Loughborough, UK) containing 9 mL of brain heart infusion (BHI) (Difco, Becton Dickinson, Heidelberg, Germany) were used to evaluate the contamination in the PIVC area. Four BHI plates with a diameter of 55 mm were positioned in front of the CV and SV areas, irrespective of the PIVC placement site (Figure 1b). The plates were numbered and positioned at the closest possible radius from the CV and SV while taking care to avoid any direct contact with the limb to prevent hair coat contamination (Figure 1b). All sedimentation plates were opened at the start of the scaling procedures. Immediately afterwards, plates number 1 (CV) and number 5 (SV) were closed (T0; baseline). Subsequently, plates 2 and 6, 3 and 7, and 4 and 8 were closed five (T5) and ten minutes (T10) into the scaling procedure and at the end (TEND) of the scaling procedure, respectively (Figure 1b).

#### 2.3.2. Part B: The Dorsal PIVC Injection Port—Sampling Procedures

The dorsal PIVC injection port was sampled with swabs just before the start of (baseline) and at the end of the dental scaling (endpoint). The cap of the dorsal PIVC injection port was manually opened. The swab from a sterile transport swab kit (Transwab, Medical Wire, Corsham, UK) was inserted into the injection port and swirled in circular movements. Thereafter, the external part of the injection port was sampled with the same swab. The investigator wore clean gloves. The swab was then rolled onto CBA plates (Becton Dickinson, Heidelberg, Germany). Each manipulation between sampling (during dental scaling) at the dorsal PIVC injection port was recorded.

### 2.4. Microbiologic Procedures

#### 2.4.1. Part A: Sedimentation Plates in the CV and SV Areas

To analyse the bacterial load resulting from airborne contamination of the PIVC area, irrespective of the PIVC treatment group allocation, BHI agar plates containing 9 mL of agar were diluted to a total volume of 20 mL with BHI broth, and serial subsequent dilutions (10^−1^, 10^−2^, and 10^−3^) were performed. The resulting dilutions were plated onto three different media: (1) CBA, clindamycin blood agar, selective for growing Gram-negative bacteria; (2) CNA, Columbia naladixicacid agar, selective for growing Gram-positive bacteria (Becton Dickinson, Heidelberg, Germany); and (3) MacConkey agar (Becton Dickinson, Heidelberg, Germany), selective for coliform bacteria, such as E. coli. Bacterial growth was determined after 48 h of aerobic incubation at 37 °C. The CFU on the plates were counted, and the total bacterial load (CFU/mL), along with the absolute CFU number, was calculated.

#### 2.4.2. Part B: The Dorsal PIVC Injection Port

The CBA plate samples obtained from the dorsal PIVC injection port were incubated aerobically at 37 °C for 48 h. After incubation, the number of CFU was determined, and single pure colonies were subjected to MALDI-TOF MS (MBT Compass Explorer, Server database: 4.1.60 (PYTH) 28 2016-04-18_11-26-19; Bruker software, Bruker Daltonics, Bremen, Germany) for identification. Colonies were extracted using a protein extraction technique, as described before [12].

### 2.5. Statistics

From the preliminary results from an analysis of 20 individuals, we estimated a ratio of the difference in the means to the standard deviation of about 0.93 for the log-transformed CFU counts between the CV and SV areas on the CBA plates and a similar value for the CNA plates. To obtain a power of 0.8 at an alpha of 0.05, about 20 individuals per group were needed for a two-sample *t*-test (which fit the design for part B of the study). The target variables were the bacterial CFU in the CBA, CNA, and MacConkey agar plates, as well as the injection port swabs. The raw data were transformed by log transformation after adding one. The bacterial contamination immediately before the start of dental scaling was compared to the bacterial loads at later time points. Statistical analysis was performed using the R software (R software for statistical computing version 3.5.3, R core team 2017, Vienna, Austria). Various explanatory variables were used. Descriptive statistics and linear models (regressions and ANOVAs) were calculated. All the *p*-values were two-sided, and *p* < 0.05 was considered statistically significant. Note that differences due to the explanatory variables were expected to increase with time such that later time points were preferable for the analyses. Due to the nature of the procedure, contamination was not expected to increase linearly with time. Due to the variability in the end time point, time point T10 was mainly tested. For each of the four conditions, CNA vs. CBA plates and CV vs. SV areas, ANOVA with the time point as a factor was performed with the log10 values as the target variables and using Tukey’s HSD test to correct for multiple testing. Mainly, we investigated whether there was a difference between the bacterial counts measured in the CV and SV areas and whether the difference was a simple function of the distance. For this purpose, data from the CV and the SV were concatenated, and the distance from the nose to the CV and SV, the length of the animal, and the severity of periodontal disease were included as covariates, respectively. Differences in the CFU counts in the CBA plates between the CV and SV areas were analysed with a linear model for each time point. As explanatory variables, the distance between the nose and the plate as well as the severity of periodontal disease were included as covariates, and the limb (the CV vs. the SV) and the individual dog were included as fixed factors. (Note that the latter are equivalent to a varying intercept random effects model.) The adjusted R-squared was used to differentiate among the fits of the models. We note that rounding may lead to *p*-values equal to zero at the reported precision, which is indicated with *p* ≈ 0. For analysis of the dorsal PIVC injection port, the frequencies of the different bacteria identified via MALDI-TOF MS analysis are reported.

## 3. Results

This study was conducted between January and October 2018. A total of 51 animals were enrolled in the study, of whom 8 were excluded for technical reasons: either the PIVC could not be placed on the first try or there were scheduling problems and the animal was not the first patient of the day in the theater (Appendix A). Furthermore, of the remaining 43 animals, 4 different animals in part A (PIVC area contamination) and 4 different individuals in part B (dorsal PIVC injection port contamination) had to be excluded due to contamination of the control plates. Finally, for part A of the study, data from 39 animals were used to compare the cephalic versus saphenous vein area contamination at four time points, regardless of the PIVC placement site (CV*n* = 18; SV*n* = 21). For part B of the study, two swabs from each animal from the cephalic vein (CV*n* = 19) and the saphenous vein (SV*n* = 20) were analysed. Dogs from nineteen breeds were included (Table 1). Their mean age was 113 ± 40 months, and their body weights ranged between 5.5 and 38.5 kg (17.18 ± 10.30 kg). The distance from the nose to the base of the tail ranged from 50 to 115 cm (80.05 ± 18.15 cm); the distances from the nose to the CV and SV areas ranged from 36.18 ± 9.9 to 79.26 ± 18.44 cm, respectively.

### 3.1. Part A: Bacterial Contamination in the CV and SV Areas

The PIVC areas at the CV and SV showed high contamination. The CFU in the CBA plates increased significantly over time, both at the CV and SV areas (CV-T5 = 2015.8974 ± 4655.2612; SV-T5 = 11.7949 ± 20.2448 CFU; CV-T10 = 32,977.4359 ± 148,770.7970; SV-T10 = 58.7179 ± 128.0456 CFU; CV-TEND = 25,528.9474 ± 42,158.6675; SV-TEND = 53.8462 ± 91.1510 CFU, data presented as mean ± standard deviation). In Figure 2, the CFU counted from the CBA and CNA plates at different time points for the CV and SV areas are also presented on a logarithmic scale. In Table 2, the pairwise differences in the log10-transformed CFU between the time points are presented for both the CV and SV areas in different (CBA and CNA) selective media with the *p*-values according to Tukey’s HSD test. The bacterial load was significantly higher at the CV compared to the SV at CBA-T5 (*p* ≈ 0), CBA-T10 (*p* ≈ 0), and CBA-TEND (*p* ≈ 0) and almost identical for the CNA plates, at CNA-T5 (*p* ≈ 0), CNA-T10 (*p* ≈ 0), and CNA-TEND (*p* ≈ 0) (also see Figure 2).

In all the results reported in this paragraph, the individual was included as a factor, which corresponds to a variable intercept random effects model, and the log-transformed CFU counts were used as the target variables. For the CBA plates, inclusion of the plaque score lowered the adjusted R-squared compared to the corresponding models where it was not included. Irrespective of the other variables included in the model, the influence of the factor limb (CV vs. SV) was always significant (*p* ≈ 0), and a model with this factor only (in addition to individual) had a relatively high R-squared (R2CBA T10 = 0.6647). When the distance from the mouth was also included, the R-squared improved slightly (R2CBA T10 = 0.6846), but the influence of the distance was only marginally significant (*p* = 0.0822). On the other hand, when the limb factor was excluded, the influence of the distance was significant (*p* ≈ 0), but the R-squared was much lower than when the limb factor was included (R2CBA T10 = 0.4734). For the CNA plates, the adjusted R-squared and significance showed identical qualitative patterns for the target variable CFU as in the CBA plates. In summary, the limb factor (CV vs. SV) explained nearly all the variation in the CFU counts for both the CBA and can plates at T10 after the start of the procedure, with the covariate of the distance from the mouth explaining very little additional variation and showing only a marginally significant influence (Figure 3). The analyses at the other time points (T5 and TEND) showed qualitatively similar but less pronounced results.

### 3.2. Part B: Bacterial Contamination and Genera Isolated from the Dorsal PIVC Injection Ports

Compared to the contamination of the CV and SV areas in part A of the study, the swabbing of the dorsal PIVC injection port in the CV (CV*n* = 19) and the SV (SV*n* = 20) showed much lower contamination (CV-Start = 5.70 ± 9.02, SV-Start = 4.5 ± 8.45; CV-TEND = 4.23 ± 7.71; SV-TEND = 6.68 ± 16.43 CFU; mean ± standard deviation, respectively). There was no significant difference in the contamination of the dorsal PIVC injection ports between the CV and the SV (*p* = 0.299) at the TEND time point. Moreover, no significant difference in the CFU counts between the start and end of the procedure was observed (*p* = 0.576). The injection port was manipulated in 14 of 39 animals. Only in two animals was the dorsal PIVC injection port manipulated more than once during dental scaling (e.g., between sampling). There was no significant influence of manipulations during the procedure on the contamination of the injection ports (*p* = 0.212).

The most commonly recovered bacteria at the injection ports belonged to the genera Staphylococcus, Neisseria, Bacillus, and Micrococcus. Staphylococcus species were retrieved from 64.2% and 41.1% of the total swab samples in the CV and SV groups, respectively, while Neisseria, Bacillus, and Micrococcus species were each found in 35.7% and 17.6%, 35.7% and 17.6%, and 21.4% and 29.4% of the total number of sampled CV and SV injection ports, respectively. No Escherichia coli or other coliforms were recovered from the PIVC injection port samples. Of all the bacteria sampled at the dorsal PIVC injection ports, 51% exhibited the growth of bacteria that could not be identified using the MALDI-TOF MS library (Figure 4).

## 4. Discussion

In this study, we compared the contamination of the CV and SV placement sites in dogs undergoing dental scaling. The results revealed significantly higher contamination in the CV area, indicating a greater contamination burden compared to that of the SV area. Furthermore, the capped dorsal PIVC injection port seems to have prevented excessive contamination. The most commonly recovered bacteria at the dorsal PIVC injection ports were identified as belonging to the genera Staphylococcus, Neisseria, Bacillus, and Micrococcus.

The production of splatter and aerosol during dental scaling is inevitable, and airborne contamination of the PIVC is likely. The plaque released during dental scaling consists of adherent opportunistic microorganisms, a matrix of shed epithelial cells, leukocytes, macrophages, blood, and saliva [3]. Considering that extra-luminal contamination is the most common pathway of PIVC contamination in human medicine [13,14], the goal of this study was to find a safe PIVC placement site for dental scaling in dogs.

Placing the sedimentation plates next to the CV and SV areas, regardless of the PIVC placement site, allowed us to compare the contamination of the two sites for each dog, accounting for the unique characteristic of each subject (Figure 2). By analysing the contamination at both sites in each dog under the same conditions, we found that the CV area carried a higher contamination burden. This area is closer to the source of aerosol production, and a higher contamination load compared to the SV was expected. Intriguingly, distance alone cannot explain the observed contamination pattern. Rather, the factor of placement position (the CV vs. SV) explained nearly all of the variability, with the influence of distance insignificant when it was additionally included.

A possible explanation for the stronger influence of the CV site rather than linear distance from the mouth could be the behavior of the two different types of airborne spray freed during dental scaling: large droplets (splatter) and aerosol. Large droplets are particles greater than 20 µm in size that fall mostly under the influence of gravity [15]. Therefore, their fall-out area might be mainly related to distance, becoming the major source of contamination in close vicinity to the oral cavity. On the other hand, smaller particles follow airflow streamlines and are potentially capable of short- and long-range transmissions. Their movement can be erratic and is largely influenced by the ambient airflow (e.g., opening of the door, the movement of the staff, etc.) but possibly also by the body shape of the patient itself.

In this study, the ambient conditions were not controlled, except that the procedures were scheduled for the early morning, so the dog was the first patient in the theatre, and no air-warming devices were used during the experimental period. The dogs were placed on a heated table or on a heated isolated electrical blanket instead. We postulate that the contamination in the CV area is mainly due to splatter, while the area of the SV is mainly contaminated due to aerosol. Dogs less than 50 cm in length from their nose to the base of their tail were excluded from the study because of possible cross-contamination of the Petri dishes in the SV area with the hair coat of the thoracic limb. The smallest dogs in our study were pugs and French bulldogs. Thus, whether the same pattern would have been observed with shorter dogs as well remains an open question.

Because the veterinary literature on our topic is sparse, we looked at several human studies of airborne contamination during periodontal treatment and its spreading within the surroundings. For instance, Timmerman et al. (2004) investigated airborne contamination at different time points using blood agar plates positioned at 40 and 150 cm only in order to compare two different aerosol suctioning devices. In another study, the contamination spread in the dental theatre was investigated by using fluorescein dye and a mannequin with phantom jaws and an evacuation device [2]. Similar to our study, Veena et al. 2015 also reported that the spread is not linear and that some areas, which lie at the same distance from the mouth, differ in contamination load. From the results of these studies, including ours, it is evident that contamination of the surroundings is inevitable, but the spread is not easy to predict and is affected by multiple factors, such as the nature of the aerosol, ambient air movement, and the positioning of the operator and the assistant [1,2,3]. The data presented here show heavy contamination of the surroundings and suggest that the use of suctioning or evacuation devices, not yet standard in veterinary medicine, might reduce the exposure of the PIVC area.

Using selective and differential growth media, we found a high bacterial load of Gram-positive bacteria, which is in line with the results from Elliot at al. (2005), who also found a significant population of Gram-positives isolated from the plaque and saliva of dogs. Although many Gram-positive bacteria are non-pathogenic, some genera, such as Staphylococcus and Streptococcus, include pathogenic species (e.g., Staphylococcus aureus and Streptococcus canis, respectively), which can cause serious health problems.

The skin flora are known to be a potential major pathway for PIVC contamination [14,16]. In dogs, the CV and SV areas harbor different skin flora. The bacterial flora at the CV site are more exposed to the saliva and food contamination. Conversely, the SV area is closer to the perineal area, making it potentially more exposed to faecal coliform contamination. Interestingly, previous studies have reported that the location (the CV vs. SV) of the PIVC is not associated with PIVC bacterial colonisation in hospitalised patients [6,8]. In the context of this study, with anaesthetised patients with PIVCs with very short dwelling times, it is unlikely that the existing skin flora would play a significant role. We found a similar composition of the bacterial flora in the CV and SV areas, as well as in the dorsal PIVC injection ports. This study was designed to look for differences in contamination between the CV and SV areas during dental scaling, and our findings highlight the importance of considering the PIVC placement site to minimise the contamination risk.

The second major pathway of bloodstream infection is contamination of the injection port [14]. In this study, the distal PIVC injection port was used only immediately after placing the PIVC when the extension line was mounted and was not sampled. However, we sampled the dorsal PIVC injection port to investigate the influence of dental scaling and manipulations on the contamination load. Our study’s results confirm that closing the dorsal PIVC injection port cap during dental scaling results in a low contamination load, regardless of the insertion site (the CV or SV). Additionally, manipulation did not influence the contamination of the injection port. This supports the findings of Seguela and Pages (2011), who studied the contamination of the PIVC tip in a clinical setting and could also not confirm a correlation between manipulation and contamination of the PIVC. The authors acknowledge the possibility that the duration of the experiment may have been too short, as well as the number of manipulations too low, to exert a significant influence.

During dental scaling, contamination of the CV and SV areas or the dorsal PIVC injection port is inevitable, but this does not automatically implicate PIVC colonisation. To confirm a causal relationship with catheter-related bloodstream infection, the same organisms would need to be recovered from the blood and the PIVC [14,16]. Interestingly, dental scaling has been associated with transient bacteraemia both in humans and dogs, which could potentially colonise the PIVC via the bloodstream [17,18,19]. Dental patients are often geriatric, with concomitant diseases and, together with immunosuppressed patients, are at a greater risk of PIVC-related infections [20,21]. The presence of transient bacteraemia during dental scaling itself suggests that it would be advisable to avoid any further bacterial challenge.

This study utilised initial dental scaling as part of its experimental design because this procedural step precedes other examination procedures and is consistent for every dental patient. Initial scaling is typically performed to improve the visibility of the tooth surfaces and gingiva, but it is undertaken before radiographs are taken and the final staging of periodontal disease is determined, along with any subsequent treatment. This represents a limitation of our study because the grading of periodontal disease was conducted before the initial scaling and radiographs, which are essential for a comprehensive assessment of the disease. The severity of periodontal disease was evaluated based on the condition of the gingiva, the presence of calculus and plaque, furcation involvement, and tooth mobility, as assessed by the attending dental surgeon. The results from this study found no correlation between the contamination load in the CV and SV areas and periodontal disease, which aligns with the findings by Nieves et al. (1997), who also showed that bacteraemia associated with dental scaling in dogs does not correlate with the degree of dental disease [18]. However, as the extra-luminal route is recognised as a major contamination pathway, the heavy bacterial load found in the PIVC area during dental scaling should already be regarded as a serious risk.

In this regard, measures to avoid bacterial contamination of the PIVC during dental procedures need to be implemented as part of good clinical practice. This includes the already common practice of a chlorhexidine oral rinse before starting the procedure and, according to the results of this study, placing the PIVC in the hind limb. Additionally, the use of suctioning devices, as employed in human medicine, could further reduce the amount of spray and aerosol released into the surrounding area. Capping the dorsal PIVC injection port also seems to have protected it from excessive contamination. Further protecting the catheter area with a bandage could be an additional strategy; however, maintaining free access to the PIVC is imperative, due to patient safety issues.

The identification of the bacteria isolated from the dorsal PIVC injection ports was conducted by MALDI-TOF MS. Staphylococcus was found in the CV- and SV-placed PIVCs in 57.9% of the samples. This corresponds to the findings of other studies that have shown various species of the genus Staphylococcus to be a part of the oral microbiota and skin flora of dogs. But it was also found on the hands of clinical staff and thus represents a major concern for catheter-related systemic infections [13,14,16,17,18,22]. This study also found Bacillus and Micrococcus species in around 27.5% and 27.2% of the samples, respectively. Furthermore, Neisseria spp., especially non-pathogenic Neisseria zoodegmatis, was found in 24.5% of the samples. Approximately 50% of the bacteria could not be identified by MALDI-TOF MS. It is highly probable that they represent previously undescribed or rare bacterial species. Our findings are supported by previous studies reporting that a large number of the bacteria within the oral microbiota in dogs are still unknown [22,23,24]. The bacterial genera described in the aforementioned studies coincide with our results. Although some pathogenic and opportunistic pathogens were found in our samples, it is not clear whether they present a significant risk of clinical infection. As the quantification of bacteria was the major aim of our study, in the current experimental setting, we used a culture-dependent approach and not DNA sequencing techniques.

The selection of the PIVC insertion site depends on various factors, including vessel availability and the purpose of catheterisation. The cephalic vein is often chosen due to its accessibility, while the saphenous vein, being shorter and more mobile, presents a greater challenge for catheterisation [25]. However, the saphenous vein is now commonly used in dogs as an alternative [26]. Through quantitative and qualitative investigations, this study highlights that PIVC area contamination should be considered a major factor for choosing the PIVC insertion site in dental patients.

## 5. Conclusions

Our data indicate significantly higher contamination of the cephalic vein area compared to the saphenous vein area, regardless of the dog’s size. Consequently, for dental scaling in dogs, it is advisable to place the peripheral intravenous catheter in the pelvic limb rather than the front limb to minimise the risk of contamination. The significantly higher bacterial load in the cephalic vein area suggests an increased risk of catheter-related bloodstream infections. Future studies are needed to investigate this potential risk and explore preventive measures.

## Figures and Tables

**Figure 1 vetsci-11-00407-f001:**
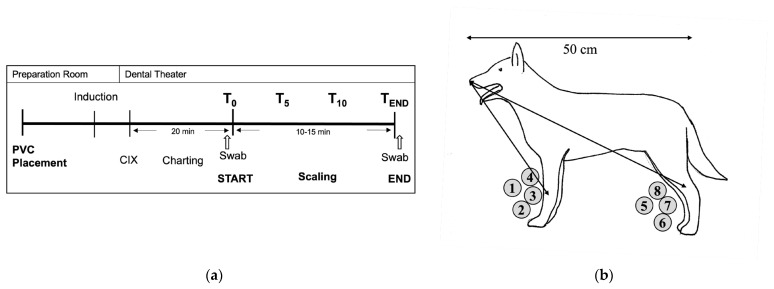
(**a**) Study protocol timeline. CIX: chlorhexidine; Induction: induction of general anaesthesia; T_0_, T_5_, T_10_, T_END_: time points for closure of sedimentation plates (study part A); Swab: time of dorsal peripheral intravenous catheter (PIVC) injection port swabbing (start and the end of the procedure, study part B). (**b**) Schematic drawing of the experimental setup of a dog positioned for dental scaling: the arrows show how the distance from the nose to the PIVC placement sites was measured for the cephalic vein (CV) and the saphenous vein (SV) and the positions of the sedimentation plates (grey and labelled 1–8); double-headed arrow above the dog shows how was the length of the dog measured (from nose to base of the tail).

**Figure 2 vetsci-11-00407-f002:**
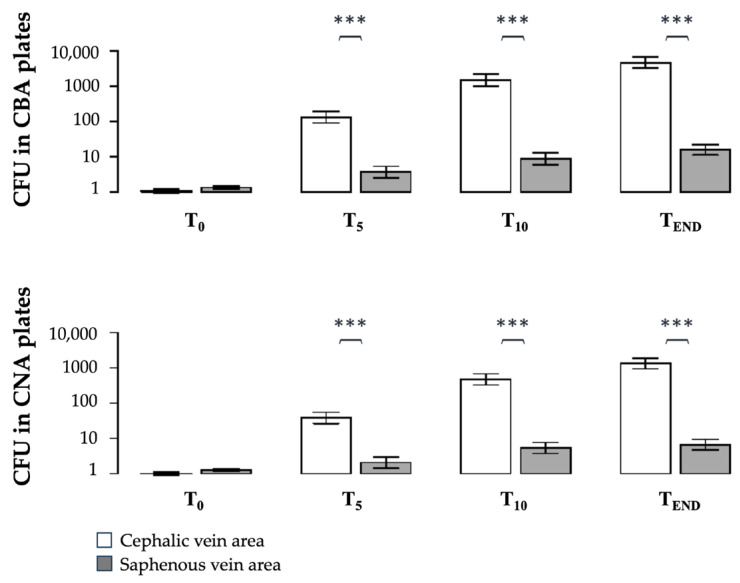
Results of part A. Number of colony-forming units (CFU) per time point (T_0_, T_5_, T_10_, T_END_) on CBA (clindamycin blood agar, selective for Gram-negative bacteria) and CNA (Columbia naladixicacid agar, selective for the growth of Gram-positive bacteria) at the cephalic vein (CV) (light columns) and the saphenous vein (SV) (dark columns), measured during dental scaling in dogs (*n =* 39). Data are presented as mean ± standard error of the log-transformed data. The average of the CFU counted per time point and allocation (CV vs. SV) was used for the statistical analysis (one-way ANOVA). Significance was set at *p* < 0.05. *** CFU differ significantly (*p* < 0.0001) between the CV and the SV at the measured time point.

**Figure 3 vetsci-11-00407-f003:**
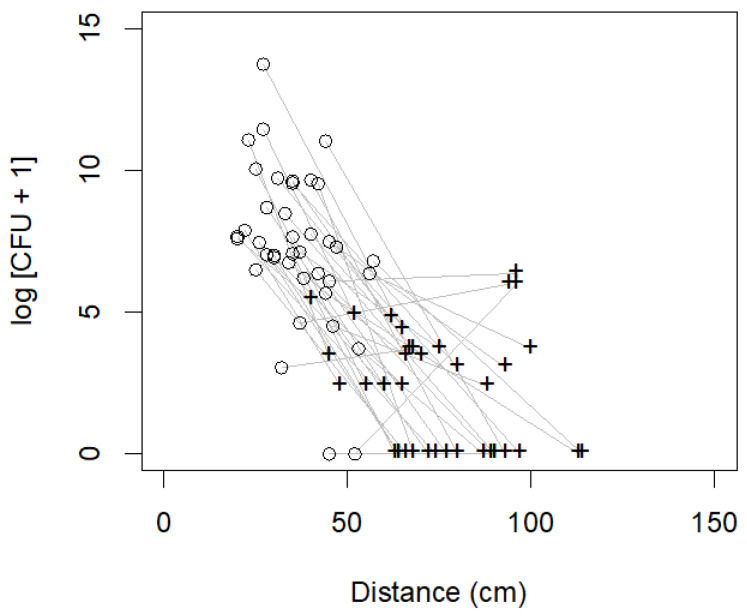
Results part A. Scatterplot of distance from the nose to the cephalic (open circles) and saphenous vein (crosses) areas of the dogs (*n* = 39) during dental scaling on the x-axis and bacterial colony-forming unit (CFU) counts of the CBA (Gram-negative-selective agar) plates on the y-axis at time point 10 (10 min into dental scaling); grey lines connect the two samples belonging to the same dog. With an increasing distance from the mouth, the CFU counts decreased (T10), except in 4 animals.

**Figure 4 vetsci-11-00407-f004:**
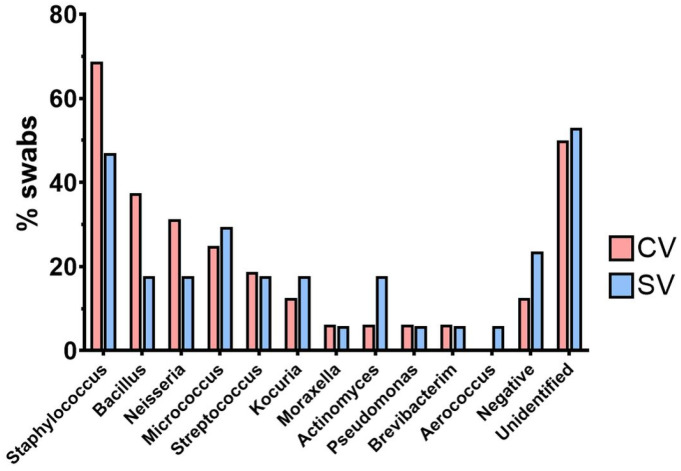
Results of part B. MALDI-TOF-MS analysis of the most frequently isolated genera from the swabs sampled from the dorsal PIVC injection ports of the dogs during dental scaling. CV: cephalic vein (*n* = 18); SV: saphenous vein (*n* = 21); %swabs: percentage of swabs with isolated bacterial species.

**Table 1 vetsci-11-00407-t001:** Population data, breed, age (in years and months), sex (m/f), weight (kg), and ASA status (1–5) of the dogs included in the study. Excluded individual dogs for part A (*, *n* = 39) and part B (#, *n* = 39) of the study.

Nr.	Breed	Age (Months)	Sex (m/f)	Weight (kg)	ASA
1	Mix	86	m	35.00	2
2	Labrador Retriever	109	m	37.80	2
3	Mix	88	f	30.00	1
4	Mix	92	m	13.30	2
5	Mix	152	f	24.75	2
6	Mix	97	f	10.50	1
7	French Bulldog	115	m	12.60	3
8	German Shepherd	93	m	34.00	2
9	Labrador Retriever	36	f	38.50	2
10	Mix	202	m	14.00	3
11 *	Pug	73	m	7.00	2
12	Mix	138	m	5.80	3
13	Mix	157	f	8.55	3
14	Mix	132	m	16.00	1
15	Mix	78	m	25.00	3
16 #	Mix	179	m	33.40	3
17	Mix	153	f	24.00	2
18	Mix	122	f	11.60	2
19	Mix	36	f	13.30	2
20	Border Terrier	159	m	12.10	3
21	Labrador Retriever	12	f	32.80	1
22	Havanese	124	f	5.50	2
23	Havanese	96	m	3.90	3
24	Boxer	75	f	28.00	1
25 *	Boxer	75	m	30.00	1
26	English Cocker Spaniel	155	f	11.20	2
27 #	Jack Russell Terrier	96	f	5.20	2
28 *	Mix	85	m	9.8	2
29 *	Pug	87	m	9.40	2
30	Mix	156	m	8.30	2
31	Mix	104	f	15.50	2
32 #	Australian Shepherd	89	f	22.20	1
33	Cavalier King Charles Spaniel	105	m	9.00	3
34	Mix	109	f	29.20	2
35	Yorkshire Terrier	81	f	3.70	2
36	Mix	142	f	9.50	2
37 #	Poodle Dog	121	m	10.00	2
38	Parson Russell Terrier	156	f	7.30	3
39 #	Collie	124	f	22.00	2
40	Beagle	106	w	13.20	1
41	Spitz Dog	73	w	7.40	2
42	Pitbull	136	w	28.00	2
43	Mix	126	w	25.00	1

Animals excluded from part A (*) or B (#) of the study.

**Table 2 vetsci-11-00407-t002:** Results of part A. Pairwise tests for the number of colony-forming units (CFU) between different time points (T_0_, T_5_, T_10_, T_END_) on CBA and CNA plates. Data are presented as differences in the log10-transformed CFU counts and pairwise comparisons between the time points per allocation (cephalic vein (CV) and saphenous vein (SV)) in dogs (*n* = 39) during dental scaling on CBA (Gram-negative-selective) and CNA (Gram-positive-selective) plates. Significance is indicated as (-) *p* > 0.1, (.) 0.1 < *p* < 0.05, (*) 0.05 < *p* < 0.01, (**) 0.01 < *p* < 0.0001, (***) *p* < 0.0001, according to Tukey’s HSD test for the six pairwise comparisons between the four time points within each of the four conditions.

Pairwise Tests	CFU in CBA		CFU in CNA	
Time Point	CV Area	SV Area	CV Area	SV Area
T_0_–T_5_	2.09 (***)	0.43 (*)	1.58 (***)	0.21 (-)
T_0_–T_10_	3.15 (***)	0.80 (***)	2.67 (***)	0.62 (**)
T_0_–T_END_	3.63 (***)	1.07 (***)	3.12 (***)	0.72 (***)
T_5_–T_10_	1.06 (***)	0.37 (-)	1.09 (***)	0.41 (.)
T_5_–T_END_	1.54 (***)	0.63 (***)	1.54 (***)	0.51 (*)
T_10_–T_END_	0.49 (-)	0.26 (-)	0.45 (-)	0.10 (-)

## Data Availability

The data that support the findings of this study are not publicly available due to privacy and ethical restrictions. As such, data sharing is not applicable to this article.

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
