# Peer review of "Does Catheter Insertion Site Matter? Contamination of Peripheral Intravenous Catheters during Dental Scaling in Dogs"

_vetsci, 2024, doi:10.3390/vetsci11090407_

Round 1

Reviewer 1 Report

Comments and Suggestions for Authors

Dear Authors,

I read your paper with a great interest. It provides novel and important data in the topic of small animal dentistry and anaesthesia. I think your results are valuable and could serve as a background for further discussion.

My comments are presented below:

- Figure 1 presents the study protocol with a good accuracy, helping to understand the study desing

- Materials and Methods & Results: you mention that the inclusion criteria was 50 cm length of the patient but the smallest included dog was weighting 5.5 kg; it would be beneficial to list the 18 included breeds, as it is difficult to imagine 5.5 kg dog with the length of 50 cm

- in my opinion, it would also be beneficial to compare the bacterial species identified in the proximity of IV catheters with the bacterial species present within the oral cavity or on the hair around the oral cavity; I understand that it was not in the scope of this research but could be beneficial to continue the research in the future, maybe to identify which location is a greater source of contamination and if one can provide more care prior or during the procedure to avoid the risk

- line 327: please either use reference number instead of the publication year in Timmerman et al. and delete the reference number at the end of the sentence or delete the publication year and use only reference number at the end of the sentence

- lines 341 and 366: as above

Author Response

Dear Authors,

I read your paper with a great interest. It provides novel and important data in the topic of small animal dentistry and anaesthesia. I think your results are valuable and could serve as a background for further discussion.

My comments are presented below:

- Figure 1 presents the study protocol with a good accuracy, helping to understand the study desing

-Answer: Thank you

- Materials and Methods & Results: you mention that the inclusion criteria was 50 cm length of the patient but the smallest included dog was weighting 5.5 kg; it would be beneficial to list the 18 included breeds, as it is difficult to imagine 5.5 kg dog with the length of 50 cm

- Answer: Thank you for pointing this out. We agree and have amended the manuscript. A new table has been introduced, that also includes the breed of the dogs as well as age, sex, weight and ASA status. Please consider that we have measured dogs from tip of the nose to base of the tail and not just the length of the back. Like this, we have indeed measured some dogs of 5.5kg with the length of tip of the nose -  base of the tail.

- in my opinion, it would also be beneficial to compare the bacterial species identified in the proximity of IV catheters with the bacterial species present within the oral cavity or on the hair around the oral cavity; I understand that it was not in the scope of this research but could be beneficial to continue the research in the future, maybe to identify which location is a greater source of contamination and if one can provide more care prior or during the procedure to avoid the risk

- Answer: Thank you, we agree it would have been interesting to have tried to compare oral cavity microbiota with bacteria found in the proximity of IV catheter. However, this very interesting question was out of the scope of this study. The goal of this study was to examine influence of dental scaling on catheter contamination. The experimental design of this study tried to exclude all other possible influences on the contamination of the catheter. This was done by sterile preparation of the skin before placement of the peripheral intravenous catheter, procedure was scheduled as first in the operating theatre, the operating theatre was not occupied 48 h before the procedure, no air-blowing warming devices were used. Further, the time points during dental measurement from 0,5, 10 min and end of the procedure are very short and with-it contamination was measured only during scaling. We have tried as much as it is possible in the clinical context to isolate this event. In that sense your question is a very interesting topic for future studies.

- line 327: please either use reference number instead of the publication year in Timmerman et al. and delete the reference number at the end of the sentence or delete the publication year and use only reference number at the end of the sentence

-Answer: Thank you for pointing this out. We have amended as suggested.

- lines 341 and 366: as above

-Answer: Thank you for pointing this out. We have done as suggested.

Reviewer 2 Report

Comments and Suggestions for Authors

In this paper, the authors have reported significantly higher chances of bacterial contamination at the location of intravenous catheter when introduced in front leg compared to hind leg in dogs. The authors have also mentioned that this occurs regardless of dog’s size or breed (n=39). The authors are requested to generate a table of these animals reporting their breed, age, sex, weight, previous known/relevant medical history.

Has there been any case in these 39 animals or in previous literature that discuss positive effect of providing cone post dental procedure that reduces chances of bacterial infection? The authors are encouraged to discuss this aspect to support this study and measures to avoid bacterial contamination.

In this study, the bacterial growth was determined for 48 hrs. at 37 degrees Celsius. What was the control plate used in this experiment to rule out any background bacterial growth without swabbing or swabbing from a non-specific region resulting in excess growth of natural microflora in BHI agar plates? The authors are encouraged to submit images of bacterial growth plates as representative images along with relevant control plate images.  

Figure 2 contain graphs with no visible x axis line.

Figure 3 is convoluted and not easy to interpret. The authors are requested to adopt a different graphical representation to show the correlation of bacterial count and its distance from the nose. The authors can also support figure 3 with CBA plate images supporting the updated graph.

Author Response

In this paper, the authors have reported significantly higher chances of bacterial contamination at the location of intravenous catheter when introduced in front leg compared to hind leg in dogs. The authors have also mentioned that this occurs regardless of dog’s size or breed (n=39). The authors are requested to generate a table of these animals reporting their breed, age, sex, weight, previous known/relevant medical history.

-Answer: Thank you for pointing this out. We have amended the manuscript and introduced the table as suggested. Please see Table2

Has there been any case in these 39 animals or in previous literature that discuss positive effect of providing cone post dental procedure that reduces chances of bacterial infection? The authors are encouraged to discuss this aspect to support this study and measures to avoid bacterial contamination.

-Answer: Thank you for your input and for bringing this to our attention. We have amended the discussion section of the study (line 445) to address possible measures for avoiding bacterial contamination of the catheter. This study only investigated up to 20 minutes into the dental scaling, as this is the phase when splatter and aerosols are actively released into the surroundings. We agree that future studies will be necessary to determine how and if further actions during the dental procedure and hospital stay can influence the contamination of the intravenous catheter.

We are not aware of any studies that have investigated the use of the cone (Elizabethan collar) during or post-operatively in dogs. In the discussion, we also mention the use of air suctioning devices and the lack thereof in veterinary medicine (lines 337-339).

In this study, the bacterial growth was determined for 48 hrs. at 37 degrees Celsius. What was the control plate used in this experiment to rule out any background bacterial growth without swabbing or swabbing from a non-specific region resulting in excess growth of natural microflora in BHI agar plates? The authors are encouraged to submit images of bacterial growth plates as representative images along with relevant control plate images.  

-Answer: Thank you for your input. In the first part of the study which looked into the contamination of BHI-sedimentation plates in vicinity of cephalic and saphenous vein in each dog. The control BHI-sedimentation plate used were 1 and 5 as on the Figure 1b. These plates were opened and immediately closed just before dental scaling started (Line 135: Immediately after, the plates number 1 (CV) and number 5 (SV) were closed (T0; baseline).

 For the second part of the study which investigated the contamination and qualitative analysis of the bacteria on the dorsal intravenous catheter injection port, the swabs were taken only from targeted areas (dorsal injection port) as this was our area of the interest. This is common practice when using sterile swabs and handling and probing in the sterile matter.

Photos of the bacterial plates are usually made for qualitative studies related to colony morphology. In this study photos of the enumerated agar plates are not available as this was a quantitative study which didn’t look into colony morphology (colony identification was performed with MALDI-TOF MS). For the first part of the study we have counted CFU from 39x3x4x2=936 plates ((39 animals, for each BHI sedimentation probe 3 different mediums (CBA, CNA and MacConkey agar) were tested per time point 0,5,10 and T-end of the procedure(x4), and each animal hat two set of probes (cephalic and saphenous are)(x2)). Plates were not photographed in the systemic manner as there were too many plates and because the morphology of the colonies was not the focus of the study. Therefore, after the enumeration and the documentation of the colonies, the plates were usually discarded. We have some photos, but we believe that presenting photo of one single plate would not benefit the reader.

Figure 2 contain graphs with no visible x axis line.

Answer: Yes, this was intended because 2 reasons: 1) a line would suggest that the left pair of bars (white) is measured before the right (grey) and 2) distances between T10 and T-end is not linear.

Figure 3 is convoluted and not easy to interpret. The authors are requested to adopt a different graphical representation to show the correlation of bacterial count and its distance from the nose. The authors can also support figure 3 with CBA plate images supporting the updated graph.

Answer: IC: Part 1: Thank you. We agree that this scatterplot is not easy to be interpreted on the first hand. We superimposed scatterplots to show the connection between contamination of the cephalic and saphenous vein area and the distance of PIVC on the cephalic and saphenous vein from the nose (nose - representing source of splatter and aerosol). We have amended the legend: Figure 3. Scatterplot of distance from the nose to cephalic (open circles) and saphenous vein area (crosses) on the x-axis and bacterial colony forming units (CFU) counts of the CBA (gram-negative selective agar) plates on the y-axis at time point 10 (10 minutes into dental scaling); a line connects two samples belonging to the same dog.

We would hope that this clarifies the problems of Reviewer 2. Alternatively, we could provide two scatterplots, one for the thoracic limb and one for the pelvic limb; however the correspondence of within the same dog would then be lost, which we consider the most interesting part of this Figure.

Part 2 of the Question: The authors can also support figure 3 with CBA plate images supporting the updated graph. Thank you for the suggestion. We have answered the request for pictures of plates above.

Reviewer 3 Report

Comments and Suggestions for Authors

The theme of the article is quite innovative and even very interesting for the development of new studies.

Line 57: a justification must be given for this high percentage in veterinary medicine

Line 70: must specifically state what type of dental procedure

Line 123: Failure to use oral radiography to objectively classify the stage of the disease should be seen as a limitation

Line 168: they state that the calculated n necessary is 40 animals but in the end they only have 39

Lines 197-206: should be in materials and methods

Figure 2: CNA and BSA without legend, as well as table 1 and figure 3.

Figure 4: refer to what CV and SV are.

Line 381: not included in the results. Will there be a difference between the degrees of severity of the disease?

Line 383: need to add comma

A limitations section must be considered, particularly in relation to the sample n.

The fact that the higher bacterial load found could result in a significant increase in the risk of bacteremia and, consequently, other clinical complications is not yet certain, as there was no evidence of this in specific clinical cases. Therefore, I believe that this should be considered in future perspectives.

The conclusion must be complemented with future perspectives.

Author Response

The theme of the article is quite innovative and even very interesting for the development of new studies.

Line 57: a justification must be given for this high percentage in veterinary medicine

Answer: The reported percentage of microbial colonisation of the peripheral intravenous catheters is confirmed by references. In the introduction we presented problems which led us to develop the study question. The line 66, a possible justification of the reported prevalence is presented/amended: “The reported prevalence of microbial colonization of peripheral PIVCs in hospitalized veterinary patients is higher than in humans and ranges between 10 - 23.2% [4–9].  This may be due to the lack of good practice guidelines for the best PIVC placement site.” Results of this study could in future be helpful when establishing guidelines for PIVC placement in this particular setting/species/group of patients.

Line 70: must specifically state what type of dental procedure

Answer: Thank you for your comment. This is very interesting and important question. In the 69 we state: “Fifty-one client–owned dogs presented for elective dental procedure were recruited.” Very often full extension of the dental disease can only be determined under general anaesthesia, which allows full clinical examination of the oral cavity. But even more often, the full extent of the dental disease, can be done, after the dental scaling and X-rays is done. This is why we chose that specific part of the whole procedure, because it precedes all others, it is always the same no matter what procedures fallow it. 2013/2020 AAHA states: “The dental- procedure involves both an awake component and an anesthetized component for a complete evaluation. It is not until the oral radiographs have been evaluated that a full treatment plan including costs of the anticipated procedure(s) can be successfully made with any degree of accuracy.”

Line 123: Failure to use oral radiography to objectively classify the stage of the disease should be seen as a limitation.

Answer: Thank you for this observation. We have amended the paragraph and added this limitation: “This study utilized initial dental scaling as part of its experimental design because this step precedes other examination procedures and is consistent for every patient. Initial scaling is typically performed to improve visibility of the teeth surfaces and gingiva, but it is done before radiographs are taken and the final staging of periodontal disease is determined, along with any subsequent treatment plans. This represents a limitation of our study because the grading of periodontal disease was conducted before the initial scaling and radiographs, which are essential for a comprehensive assessment of the disease. The severity of periodontal disease was evaluated based on the condition of the gingiva, the presence of calculus and plaque, furcation involvement, and tooth mobility, as assessed by the attending dental surgeon. Results from this study found no correlation between contamination load in the CV and SV area and the of the periodontal disease which aligns with findings by Nieves et al. (1997) who also showed that bacteraemia associated with dental scaling in dogs does not correlate with the degree of dental disease [19]. However, as the extra-luminal route is recognised as a major contamination pathway, the heavy bacterial load found in the PIVC area during dental scaling should already be regarded as a serious risk. “

Line 168: they state that the calculated n necessary is 40 animals but in the end they only have 39

Answer: We calculated that "approximately 40" individuals (as we report in the materials and methods section) would provide a power of at least 80% to obtain significant results at the 0.05 level. As this is a clinical study, drop-outs are to be expected. We therefore enrolled 51 rather than 40 individuals but had to exclude some. Obviously the 39 (just one less than 40, which would lower the a priori power to only slightly below 80%) finally included in the study were enough to detect significant differences. From which we concluded that the power calculation was quite successful.

Lines 197-206: should be in materials and methods

Answer: We reported the planning and methods in the materials and methods section, but the specific details of the study, which could not be planned, in the results section. This part could also be moved to the materials and methods (as the dogs would provide the "material" for the study), but since these aspects can only partly be controlled, the information could also be reported in the results section.

Figure 2: CNA and BSA without legend, as well as table 1 and figure 3.

Answer: Thank you for this observation, we have done as suggested also in other figures.

Figure 4: refer to what CV and SV are.

Answer: Thank you again. We corrected this missed information.

Line 381: not included in the results. Will there be a difference between the degrees of severity of the disease?

Answer: This study did not find the correlation between the severity of the disease and the contamination of the peripheral intravenous catheter (Results Line 249, Discussion Line 418-421).

Line 383: need to add comma

Answer: Yes, thank you. Change was made as requested.

A limitations section must be considered, particularly in relation to the sample

Answer: --See also the previous answer.

The fact that the higher bacterial load found could result in a significant increase in the risk of bacteremia and, consequently, other clinical complications is not yet certain, as there was no evidence of this in specific clinical cases. Therefore, I believe that this should be considered in future perspectives.

The conclusion must be complemented with future perspectives.

Answer: Thank you very much for this observation. We agree and have amended the conclusion as suggested: “Our data indicate significantly higher contamination of the cephalic vein area compared to the saphenous vein area, regardless of the dog’s size. Consequently, for dental scaling in dogs, it is advisable to place the peripheral intravenous catheter in the pelvic limb rather than the front limb to minimize the risk of contamination. The significantly higher bacterial load in the cephalic vein area suggests an increased risk of catheter-related blood stream infections. Future studies are needed to investigate this potential risk and explore preventive measures.”

Reviewer 4 Report

Comments and Suggestions for Authors

overall it is a interisting paper regarding contamination during dental procedures of the cateters. Some comments :

abstract: develop more why the contamination occur if the catter is alredy in place during procedure, which bacterias are more common,

methods: is not clear if the dogs are all the same size

results; subtitles are after the figures

table 1 is to large, inprove

bacteria names in latim are in italic - correct

discussion: there is not studies in other animals like rabbits?

Comments on the Quality of English Language

good

Author Response

overall it is a interisting paper regarding contamination during dental procedures of the cateters. Some comments :

abstract: develop more why the contamination occur if the catter is alredy in place during procedure, which bacterias are more common,

Answer: Amended as suggested: “Contamination of the peripheral intravenous catheter (PIVC) is unavoidable, due to splatter and aerosol generated during dental scaling in dogs under general anaesthesia.”

methods: is not clear if the dogs are all the same size

Answer: The dogs are not all the same size. The inclusion criteria are described line 71-72 (Inclusion criteria were ASA classification I to III, and length of the animal from nose to base of the tail of 50 cm or more.)

results; subtitles are after the figures

Answer: I’m very sorry, but I have not understood this request. We are happy for every chance to improve the manuscript and would be happy to make changes if necessary.

table 1 is to large, inprove

Answer: Yes, thank you. We have done as suggested.

bacteria names in latim are in italic – correct

Answer: Thank you for noticing. We have made changes as suggested.

discussion: there is not studies in other animals like rabbits?

Answer: We tried to answer one very specific question in a very complex setting consisting of:  oral microbiome, oral microbiome during dental scaling, skin microbiome, contamination of the catheter in the hospital versus contamination of the catheter during dental scaling.  To our knowledge there are no comparable studies in other animals or rabbits. Dental scaling is common procedure in cats and dogs, but not in other animals this is why probably there are no comparable studies. Rabbits usually suffer from dental plaque and calculus but more from the dental abscesses and periodontal/apical infections and so dental scaling is not a common procedure in this species.

Round 2

Reviewer 3 Report

Comments and Suggestions for Authors

Line 57: if possible, I think it would be very enriching for the work to explore in more detail, based on the bibliographical references cited, the percentages that exist in veterinary medicine.

Figure 2 is still not labelled in terms of CNA and BSA.

Author Response

Line 57: if possible, I think it would be very enriching for the work to explore in more detail, based on the bibliographical references cited, the percentages that exist in veterinary medicine.

Reply: Thank you for insisting to check this line. We are grateful for every chance to improve our paper. We have amended L57: “The reported prevalence of microbial colonization of peripheral intravenous catheters (PIVCs) in hospitalized veterinary patients ranges between 10.4% and 39.6% [4–9]. However, veterinary medicine currently lacks comprehensive guidelines for identifying the best PIVC placement site, which could help reduce contamination and improve patient outcomes.”

References listed:  overall prevalence of microbial colonisation of the peripheral venous catheter (xx)

  1. Jones, I.D.; Case, A.M.; Stevens, K.B.; Boag, A.; Rycroft, A.N. Factors Contributing to the Contamination of Peripheral Intravenous Catheters in Dogs and Cats. Vet. Rec. 2009, 164, 616, doi:10.1136/vr.164.20.616. 20% (23 of 99)
  2. Mathews, K.A.; Brooks, M.J.; Valliant, A.E. A Prospective Study Of Intravenous Catheter Contamination. J. Vet. Emerg. Crit. Care 1996, 6, 33–43, doi:10.1111/j.1476-4431.1996.tb00032.x. 10,7%
  3. Ramos, P.J.G.; Pérez, C.F.; Santiago, T.A.; Artigao, M.R.B.; Ortiz‐Díez, G. Incidence of and Associated Factors for Bacterial Colonization of Intravenous Catheters Removed from Dogs in Response to Clinical Complications. J. Vet. Intern. Med. 2018, 32, 1084–1091, doi:10.1111/jvim.15118. 39,6%
  4. Seguela, J.; Pages, J. ‐P. Bacterial and Fungal Colonisation of Peripheral Intravenous Catheters in Dogs and Cats. J. Small Anim. Pr. 2011, 52, 531–535, doi:10.1111/j.1748-5827.2011.01101.x. 15·4%
  5. Marsh-Ng, M.L.; Burney, D.P.; Garcia, J. Surveillance of Infections Associated With Intravenous Catheters in Dogs and Cats in an Intensive Care Unit. J. Am. Anim. Hosp. Assoc. 2014, 43, 13–20, doi:10.5326/0430013. 24,5%
  6. Matula, E.; Mastrocco, A.; Prittie, J.; Weltman, J.; Keyserling, C. Microorganism Colonization of Peripheral Venous Catheters in a Small Animal Clinical Setting. J. Vet. Emerg. Crit. Care 2023, 33, 509–519, doi:10.1111/vec.13328. 10,4%

Figure 2 is still not labelled in terms of CNA and BSA.

Thank you, we amanded as suggested